# PERCEPTION THROUGH SPARSITY: FUSING AND ENHANCING MULTI-AGENT SPARSE REPRESENTATION WITH DEFORMABLE CROSS-ATTENTION

## ABSTRACT

Multi-agent perception has gained significant attention for its ability to share information among connected automated vehicles (CAVs) and smart infrastructure, thus mitigating occlusions and extending effective sensing range. Despite this progress, research on radar-based cooperative perception has been constrained by limited datasets, where existing benchmarks either provide only partial radar views or a small number of frames, making it difficult to fully study radar's potential in V2X perception. To address this gap, we introduce **V2XSet-R**, the first large-scale dataset that provides *complete* $360°$ *radar coverage* to expand the scale and diversity of radar data, enabling systematic study of radar-based cooperative perception and fusion. Building on this resource, we propose **SparseFusion**, a dual-stage fusion framework tailored to sparse multi-agent perception. Unlike prior position-wise self-attention designs that compute affinity scores only among voxels at the same BEV location, SparseFusion aggregates cross-voxel context via a query-based deformable attention module that adaptively samples informative regions across space and agents. This design overcomes sparsity-induced degeneration and enhances feature interaction across agents and effectively generalizes to camera BEV features. These results demonstrate that SparseFusion is a precise, efficient, and modality-agnostic fusion method for cooperative perception.

## 1 INTRODUCTION

Multi-agent perception extends the sensing range and robustness of a single autonomous vehicle by allowing multiple vehicles and roadside units to exchange perception information. Through this shared mechanism, agents can jointly reason about occluded objects, compensate for individual sensor blind spots, and improve situation awareness under diverse weather and operation conditions that degrade line of sight sensors. Early benchmarks Xu et al. (2021b); Wang et al. (2020b); Chen et al. (2019); Meng et al. (2023) quantify these benefits, showing that cooperative fusion can significantly improve 3D detection accuracy over single-agent baselines. Despite these gains, most systems still hinge on camera/LiDAR, which falter in rain, fog, glare, and at long range—precisely when collaboration matters. Accordingly, radar-centric cooperative perception is emerging as a critical cornerstone due to radar's inherent robustness to adverse weather and lighting and motion-aware sensing, making it essential for all-weather, safety-critical operation.

However, progress on radar-based cooperative perception has been constrained by the lack of high-quality benchmarks. OPV2V Xu et al. (2021b) omits radar entirely, while V2X-R Huang et al. (2024) and V2X-Radar Yang et al. (2024) include radar but equip each vehicle and roadside unit with only a single, 120° fields of view (FOVs), yielding non-holistic coverage with large azimuth gaps. The restricted FOVs and the insufficient sensor count make it difficult to systematically evaluate radar's unique advantages (e.g., long-range, Doppler-aware cues in adverse weather) and its challenges (e.g., cross-agent calibration, synchronization, and fusion for sparse returns) in multi-agent settings. To fill this gap, we introduce **V2XSet-R**, the first large-scale dataset that provides complete 360° radar coverage from both vehicles and infrastructure to increase the scale of available radar data and enables a detailed study of radar-centric cooperative perception.

At the algorithmic level, most vehicle-to-everything (V2X) fusion modules still rely on position-wise self-attention that computes affinities only between agents in the same bird's-eye-view (BEV) cell. Such designs constrain the sparse measurements BEV maps fusion, which is dominated by empty cells, yielding near-uniform attention and overlooking the cross-voxel context where the informative geometry resides. Recent cooperative perception networks such as *V2X-ViT* Xu et al. (2022),

*CoAlign* Lu et al. (2023), and *Where2Comm* Hu et al. (2022) transpose the transformer paradigm to V2X settings. After rigidly aligning every agent's BEV feature map with a shared coordinate frame, they apply self-attention to all agents at each voxel position. Formally, for every spatial index $(i,j)$ the per-agent embeddings $\left\{\mathbf{F}_{:,i,j}^{(k)}\right\}_{k=1}^{N} \subset \mathbb{R}^C$ are stacked into an $N$-token sequence and updated via multi-head attention:

$$\mathbf{Z}_{i,j} = \text{Attn}\Big(\big[\mathbf{F}_{:,i,j}^{(1)}\|\ldots\|\mathbf{F}_{:,i,j}^{(N)}\big]\Big), \qquad \mathbf{Z}_{i,j} \in \mathbb{R}^{N \times C}. \tag{1}$$

Although this strategy inherits the permutation invariance and global information exchange that make transformers effective in NLP, it exhibits two key mismatches with cooperative sensing. (1) In language models, each token already carries rich semantic content. Attention coefficients therefore encode meaningful linguistic relations. However, in V2X fusion, a token is simply an agent-specific voxel feature. The multi-agent feature fusion is computed only among agents occupying the same voxel, ignoring spatial correlations between neighboring voxels. This assumption holds for dense LiDAR grids but fails for 4D radar, whose non-zero voxels occupy only a small fraction of the BEV plane. In such sparse maps, the voxels that capture the geometry of objects are often isolated, and the meaningful context resides in the adjacent

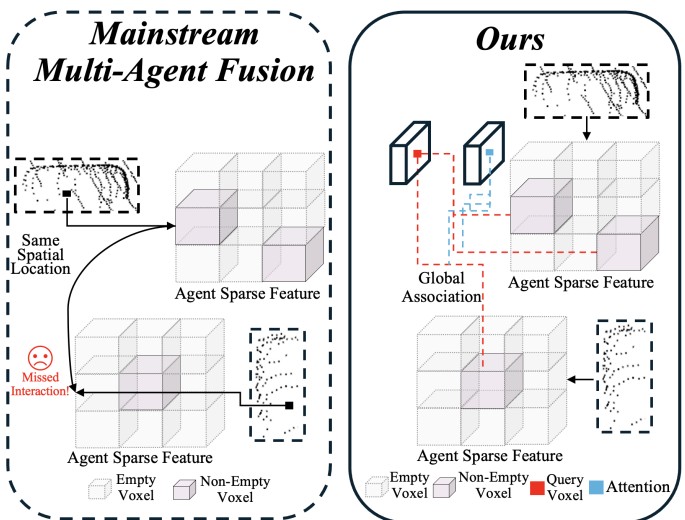

Figure 1: **The fusion method of mainstream multi-agent detectors and our SparseFusion.** Mainstream multi-agent fusion only considers the spatial relationship between the same voxels position. In contrast, SparseFusion, which leverage the query voxel to predict the offsets of the selected multi-agent features, consider the global relationships between the multi-agent sparse features.

non-empty voxels rather than in the same location across agents. Consequently, position-wise attention misses crucial geometric cues and fails to aggregate the cross-voxel evidence needed for accurate radar perception.(2) The sparse sensor measurements often collapse to the zero vector when the sensor observes nothing at $(i,j)$. Because any dot product involving a zero key or query is also zero, the scaled attention logits degenerate to a constant, and the subsequent attention map thus carries little discriminative signal that hampers effective learning.

These limitations motivate our SparseFusion design, a method crafted to tackle the sparsity challenges in multi-agent sensor measurement fusion. At its heart is a dual-fusion module that selects 3-D objects directly from voxel features. This design removes the constraints for same-spatial-location fusion, solves sparse-to-dense feature-association issues, and broadens the range of multi-agent feature interaction. Fig. 1 compares the pipeline of our approach with those of mainstream 3-D detectors. A key deficiency of mainstream V2X detectors is that their fusion stage attends only to agent features occupying the same voxel, thus overlooking informative cross-voxel context in sparse radar grids. As visualized in Fig. 1, the left panel shows that the non-empty voxel interacts with the nearby empty voxel, resulting in missed interaction. In contrast, the right panel depicts our SparseFusion design. The query-based deformable attention block predicts continuous offsets (dashed red line) that jump to the most informative neighboring voxels and generate the attention score for the predicted informative voxels, regardless of agent index or spatial coincidence. This fusion design removes the hard constraint of same-location matching, converts sparse radar evidence into dense object-centric features, and enlarges the scope of multi-agent interaction, all while keeping the communication payload fixed. Furthermore, we introduce a multi-agent feature enhancement module that leverages RoPE-augmented sparse self-attention to enrich non-empty BEV pillars with contextual cues across

agents, thereby strengthening cross-voxel interaction without increasing communication cost. Our contributions are:

• We introduce SparseFusion, which explicitly aggregates sparse multi-agent features for 3D perception and generalizes across modalities, delivering state-of-the-art performance and superior flexibility across radar and LiDAR-based cooperative perception.

• We introduce a dual-stage fusion that (i) performs lightweight spatial fusion and (ii) applies query-based deformable cross-attention to aggregate informative cross-voxel, cross-agent cues to address sparsity and the failure modes of position-wise BEV attention. Furthermore, a multi-agent feature enhancement block with RoPE-augmented sparse self-attention is proposed to densifies non-empty pillars and broaden multi-agent interactions beyond same-cell matches.

• We release the first large-scale V2X benchmark with complete 360° radar coverage from vehicles and infrastructure, enabling systematic study of radar-centric multi-agent perception. We have implemented this pipeline with various fusion strategies and provided a comprehensive benchmark on our V2XSet-R dataset, boosting the research in multi-agent fusion for cooperative 3D object detection.

## 2  RELATED WORK

### 2.1  V2X PERCEPTION METHODS

Modern V2X detectors typically extract each agent's BEV feature and then fuse shared features by using a position-wise self-attention. V2X-ViT Xu et al. (2022) is a representative example, using a vision transformer with multi-scale window attention after alignment. Where2Comm Hu et al. (2022) selects spatially critical regions via a confidence map, and How2Comm Yang et al. (2023a) jointly optimizes bandwidth/latency trade-offs under practical transmission noise. Earlier pipelines established the paradigm of intermediate feature sharing. F-Cooper Chen et al. (2019) proposed feature-level exchange to cut bandwidth. V2VNet Wang et al. (2020b) formalized end-to-end feature aggregation with learned communication. Recent work CORE Wang et al. (2023) reconstructs missing context of collaborators to improve communication efficiency.

To improve robustness, several works target misalignment, heterogeneity, and domain shifts. CoAlign Lu et al. (2023) explicitly compensates for pose errors when aligning multi-agent features. HEAL Lu et al. (2024) builds a unified feature space for heterogeneous agents and modalities with an extensible fusion interface. SCOPE Yang et al. (2023b) introduces spatio-temporal domain awareness, combining temporal cues and deformable cross-attention to be resilient to localization errors and heterogeneous agents. Recent AgentAlign Meng et al. (2024) addresses multi-modality misalignment by learning a cross-modality feature-alignment space and heterogeneous-agent alignment mechanism, and evaluates robustness under synthetic sensor noise. SiCP Qu et al. (2024) proposes a dual-perception network that supports individual and cooperative operation with lightweight coupling.

Overall, these frameworks improve communication efficiency, alignment, and heterogeneity, yet most methods still conduct attention within identical spatial cells. This fusion mechanism underutilizes the cross-voxel context and struggles on sparse BEV modalities (radar, camera). In order to address this gap, our SparseFusion proposes the dual-fusion method to fuse the nearby voxel feature of the varying number of agents and densify the sparse multi-agent feature through proposed enhancement module.

### 2.2  TRANSFORMERS FOR SPARSE DATA

The vanilla Transformer enables global interactions but scales quadratically with sequence length Vaswani et al. (2017). Sparse Transformer targets byte-level text, images, and audio, and factorizes self-attention into fixed local and strided patterns, reducing complexity to $\mathcal{O}(n\sqrt{n})$ while retaining long-range dependencies in very long sequences Child et al. (2019). Longformer addresses long-document NLP by combining a sliding window with a small set of global tokens, yielding linear complexity that matches the sparsity of informative cross-sentence links Beltagy et al. (2020). BigBird further augments local windows with random and global connections to achieve linear complexity with theoretical guarantees, suitable for extremely long NLP sequences where only a subset of token pairs must interact Zaheer et al. (2020).

Another direction approximates the attention kernel. Linformer demonstrates that attention matrices are approximately low rank and projects keys/values to a lower dimension, producing linear-time attention that works well on long text with reduced memory Wang et al. (2020a). In vision, sparsity often appears as a small number of salient spatial regions rather than long sequences. Deformable

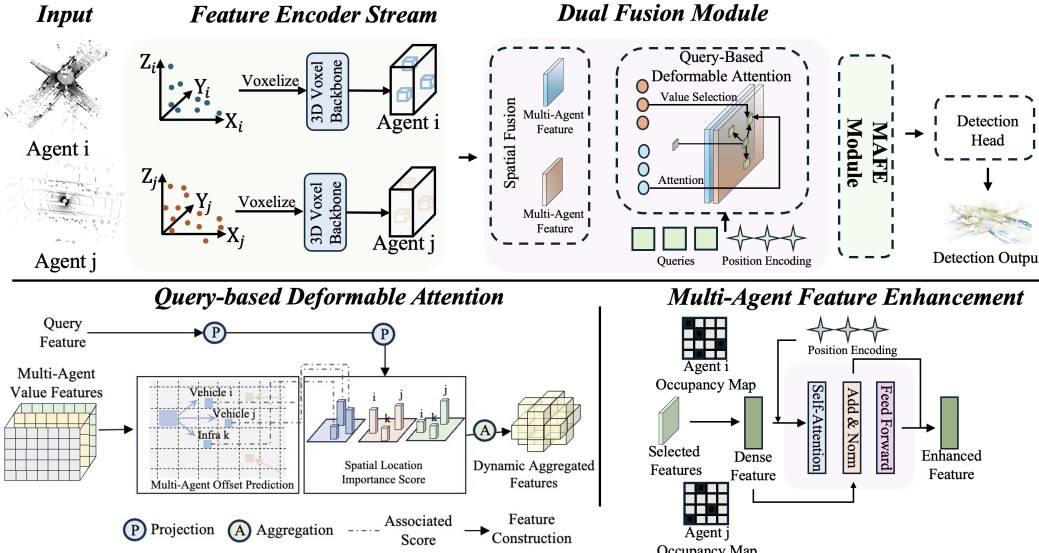

Figure 2: **The detailed structure of our proposed cooperative perception network.** The proposed SparseFusion consists of feature encoder stream, dual fusion module and detection head. Initially, multi-modality features are extracted from sensor measurements. Subsequently, the spatial alignment is used as the first stage fusion to aggregate the multi-agent features. The query-based deformable fusion is used as the second stage to select the features from the multi-agent features and enhance the feature interaction. Finally, the multi-agent feature enhancement module further enhance the fused features by injecting positional encoding and applying sparse self-attention across occupancy maps.

DETR predicts a few sampling offsets around reference points on multi-scale feature maps and aggregates only those samples, improving efficiency and convergence for object detection with sparse visual evidence Carion et al. (2020). In 3D perception, Point Transformer adapts attention to irregular and sparse point clouds by restricting interactions to local $k$-NN neighborhoods and injecting relative positional encodings, keeping cost proportional to local point density while preserving geometry Zhao et al. (2021). Our method follows this sampling philosophy: instead of exhaustive position-wise attention on BEV grids with many empty cells, we learn query offsets that select informative cross-voxel and cross-agent locations tailored to sparse radar and camera features. Despite their efficiency, these transformers either impose fixed sparsity patterns or approximate the attention kernel, or they learn sampling within a single image and point cloud. None of methods is designed for cross-agent BEV fusion dominated by empty tokens, where informative geometry lies in neighboring voxels from varying number of sources. Position-wise self-attention used in current V2X models further isolates voxels, missing the cross-voxel evidence essential for sparse radar/camera features. We address this with SparseFusion, which leverages the dual fusion module to sample the cross-voxel, cross-agent locations to aggregate the multi-agent features.

## 3 METHODOLOGY

### 3.1 OVERALL ARCHITECTURE

As introduced in Section 1, the major objective of our design is to overcome the sparsity issue encountered in the multi-agent fusion stage. Therefore, SparseFusion is designed to enhance the voxel feature interaction for the sparse and empty voxel feature. The proposed cooperative perception network is visualized in Fig. 2. First, every agent voxelises its local point cloud and transforms it into a BEV tensor by passing pillars through a Pillar VFE, a scatter operation, and a lightweight CNN backbone. Second, the BEV tensors from all agents enter the dual fusion module. A preliminary spatial fusion layer computes the spatial interaction across agents, thus producing initial multi-agent feature planes for each grid location. These planes are then refined by MFAFuser, a multi-layer transformer that leverages query-based deformable cross-attention together with learnable positional embeddings to gather the most informative features from neighboring agents while respecting spatial geometry. Finally, the fused BEV feature is forwarded to the enhancement mod-

Table 1: **Performance comparison on the V2X-R validation and testing dataset.**

| Methods | Publication | 3D mAP@Our V2XSet-R | | | 3D mAP@V2X-R | | |
|---|---|---|---|---|---|---|---|
| | | IoU=0.3 | IoU=0.5 | IoU=0.7 | IoU=0.3 | IoU=0.5 | IoU=0.7 |
| V2XViT Xu et al. (2022) | ECCV2022 | 65.8 | 64.5 | 50.2 | 80.61 | 73.52 | 42.60 |
| Where2comm Hu et al. (2022) | NeurIPS2023 | 75.3 | 73.6 | 45.3 | 79.21 | 72.88 | 36.15 |
| SCOPE Gamerdinger et al. (2024) | ICCV2023 | - | - | - | 73.00 | 71.60 | 51.60 |
| CoAlign Lu et al. (2023) | ICRA2023 | 78.0 | 75.4 | 52.9 | 81.69 | 75.74 | 52.01 |
| AdaFusion Qiao & Zulkernine (2023) | WACV2023 | 78.9 | 77.2 | 60.7 | 81.95 | 77.84 | 55.32 |
| SICP Qu et al. (2024) | IROS2024 | - | - | - | 71.94 | 65.17 | 63.44 |
| Our | - | 82.2 | 81.7 | 70.8 | 85.3 | 82.3 | 64.1 |

ule to densify the sparse multi-agent features, which is used by convolutional block and regression heads that output object heatmaps and 3-D bounding boxes.

## 3.2 FEATURE ENCODER STREAM

As shown in Fig. 2, SparseFusion begins with the feature encoder stream. Our feature extraction method is based on the PointPillar and can be generalized to other voxel-based backbone. First, synchronized measurements from multiple 4-D radars are voxelized. After voxelisation, each pillar is processed by the Pillar VFE to obtain a fixed-length representation of the contained points. These embeddings are scattered onto a two-dimensional BEV feature grid, producing a dense BEV feature map. Because the encoder operates purely on voxelised features, it can be seamlessly replaced by alternative voxel-based backbones (e.g., SECOND Yan et al. (2018), Voxelnet Zhou & Tuzel (2018)) without modifying the downstream dual-fusion stages.

## 3.3 DUAL FUSION MODULE

### 3.3.1 SPATIAL FUSION

Multi-agent perception aggregates multi-source BEV features arriving from multiple vehicles. Directly concatenating all agent maps is computationally costly and sensitive to the varying number of collaborators. We therefore introduce a global fusion stage that collapses the agent dimension by computing saliency and consensus maps over all agents. Specially, the saliency map is computed by using the max pooling which preserves the strongest, most confident response at every grid cell, whereas the consensus map leverages the mean pooling operation to capture the consensus context. The two feature maps can be obtained by the following equations:

$$\mathbf{F}_{\max}^c(i,j) = \max_{k \in \{1,\dots,N\}} \mathbf{F}_{c,i,j}^{(k)},$$

$$\mathbf{F}_{\text{mean}}^c(i,j) = \frac{1}{N} \sum_{k=1}^{N} \mathbf{F}_{c,i,j}^{(k)}, \quad (2)$$

The two statistics jointly provide a permutation-invariant summary that is lightweight to communicate and serves as an informative seed for the subsequent attention-based fusion.

### 3.3.2 QUERY-BASED DEFORMABLE ATTENTION

The generated features from global spatial fusion still cannot distinguish which neighboring agents contribute the most reliable evidence and explicitly model long-range multi-agent interactions. Therefore, we leverage the query-based Deformable Attention (Q-DA) addresses this by letting each BEV query dynamically select a sparse set of informative sampling points from the fused maps. The learned offsets steer each query toward high-value regions in both the saliency and consensus planes, enabling the model to focus on truly discriminative cues while ignoring redundant or noisy cells. Because the offsets are predicted from the query itself rather than being fixed, Q-DA can flexibly capture local neighbor multi-agent interaction by predicting small offsets and adaptively expand the receptive field to consider global context by generating large offsets. This adaptive receptive field promotes richer multi-agent interaction without exploding computational cost or bandwidth.

**BEV–query generation**   Let $\mathbf{F}_{\max}, \mathbf{F}_{\text{mean}} \in \mathbb{R}^{C \times H \times W}$ be the saliency and consensus maps from Spatial Fusion; let $W_q \in \mathbb{R}^{D \times 2C}$ be a learnable linear projection; and let $\mathbf{P} \in \mathbb{R}^{D \times H \times W}$ denote a learned 2-D positional tensor. The entire query sequence is obtained as the following step:

$$\widetilde{\mathbf{Q}} = W_q \left[ F_{\max} \,\|\, F_{\text{mean}} \right] \mathbf{P} \quad (3)$$

Here, $[\cdot \| \cdot]$ indicates channel-wise concatenation. The resulting tensor $\widetilde{\mathbf{Q}} \in \mathbb{R}^{D \times H \times W}$ is subsequently flattened into an HW-length token sequence and passed as the query input to the stacked

query-based deformable attention blocks, where each token adaptively aggregates multi-agent context before the downstream detection head.

**Sparse sampling with deformable attention**  Let $\tilde{\mathbf{q}}_i \in \mathbb{R}^D$ be the $i$-th BEV query token in $\mathbf{Q}$. For every attention head $h \in \{1, \dots, H\}$, multi-agent feature map $m$, and sampling point $p \in \{1, \dots, P\}$, the network predicts an offset $\Delta_{h,m,p,i} \in \mathbb{R}^2$, and a normalized weight $\alpha_{h,m,p,i}$ with $\sum_{m=1}^{M} \sum_{p=1}^{P} \alpha_{h,m,p,i} = 1$.

Here, the vector $\Delta_{h,m,p,i} \in \mathbb{R}^2$ stores the learnable offset relative to the cell center that determines *where* to sample, while $\alpha_{h,m,p,i} \in (0,1)$ is the corresponding importance weight, normalized such that $\sum_{m=1}^{M} \sum_{p=1}^{P} \alpha_{h,m,p,i} = 1$ for every head $h$ and query $i$.

Given the reference point of cell center $\mathbf{r}_i \in [0,1]^2$ and the size $S_m = (W_m, H_m)$ of modality $m$, the continuous sampling location is

$$\mathbf{s}_{h,m,p,i} \;=\; \mathbf{r}_i \frac{\Delta_{h,m,p,i}}{S_m}, \tag{4}$$

where the division is element-wise.

Each query then aggregates a sparse set of bilinearly interpolated values $\mathcal{V}_{m,h}(\cdot)$ drawn from the multi-agent saliency and consensus maps:

$$\widehat{\mathbf{q}}_i \;=\; \sum_{h=1}^{H} \sum_{m=1}^{2} \sum_{p=1}^{P} \alpha_{h,m,p,i} \, \mathcal{V}_{m,h}\big(\mathbf{s}_{h,m,p,i}\big). \tag{5}$$

The outputs of all heads are concatenated and projected through a linear layer $W_o \in \mathbb{R}^{D \times D}$, resulting in the refined query representation that feeds the subsequent feedforward network:

$$\mathbf{q}_i^{\text{out}} = W_o\big[\widehat{\mathbf{q}}_i^{(1)} \| \widehat{\mathbf{q}}_i^{(2)} \| \cdots \| \widehat{\mathbf{q}}_i^{(H)}\big].$$

We repeat this deformable fusion process $L$ times to produce the final BEV tensor.

### 3.4 MULTI-AGENT FEATURE ENHANCEMENT

After obtaining the multi-agent features, we merge their transformed coordinates, denoted as the agent occupancy map in 2, into a unified set that directly indicates the non-empty BEV grid positions contributed by the multi-agent features. Different from the usual BEV fusion, the merged occupancy maps only select the feature in non-empty BEV grid and yield a compact representation of all valid BEV cells. Then, to encode spatial structure, we apply a 2D Rotary Positional Embedding (RoPE), which injects continuous positional information along both horizontal and vertical grid axes. On top of these position-aware features, we perform self-attention: each non-empty pillar feature is projected into queries, keys, and values, attention scores are computed as the similarity between queries and keys, and the attended values are aggregated to enhance each pillar with contextual information from all other non-empty cells. A feed-forward network with residual connections and normalization completes the update before scattering the enhanced features back into the dense BEV grid.

**Detection Head**  The enhanced BEV tensor produced by the stacked deformable–attention layers is fed into a lightweight detection head composed of parallel convolutional branches for object classification and 3-D bounding-box regression. This head operates on the shared BEV grid, allowing every agent to output a unified set of world-aligned detections.

## 4  V2XSET-R DATASET

**Simulator and Sensor Configuration**  To generate our V2XSet-R dataset, we used OpenCDA Xu et al. (2021a) integrated with CARLA, a cooperative driving simulation platform that supports multiple agents and provides control over embedded vehicular communication protocols. This setup enabled us to simulate realistic V2X environments and collect synchronized multi-modal sensor data for both vehicles and infrastructure.

The simulation environment was equipped with a diverse set of sensors to ensure rich perception capabilities as shown in Table2. Each vehicle and infrastructure was outfitted with four RGB cameras (800×600 resolution, 110° FOV), one high-resolution LiDAR (64 channels, 120 m range, 10 Hz rotation), four radar units (150 m range, 120° horizontal, 40° vertical FOV), and GPS/IMU modules with precise positional and heading accuracy. This comprehensive sensor suite ensures robustness in perception and provides complementary modalities for cooperative tasks.

**Dataset Comparison and Advantages**   Table 3 compares V2XSet-R with existing datasets. Unlike prior benchmarks such as OPV2V, V2XSet, and V2X-R, our dataset uniquely provides 360° radar coverage and supports infrastructure-to-infrastructure (I2I) communication, which has been largely overlooked in previous works. Importantly, V2XSet-R delivers a more complete radar frame collection from both vehicles and roadside infrastructure to greatly surpassing existing datasets that only provide limited radar data or none at all.

In addition, V2XSet-R maintains large-scale coverage across modalities, with 142k radar frames, and 160k annotated 3D bounding boxes, making it well-suited for robust multi-modal and cooperative perception research. These enhancements make V2XSet-R the first dataset to fully integrate radar sensing into cooperative V2X perception tasks while maintaining scale and diversity comparable to leading benchmarks.

Table 2: **Sensor setting.**

| Sensors | Details |
|---|---|
| 4 × Camera | RGB, 800 × 600 resolution, 110° FOV |
| 1 × LiDAR | 64 channels,120m range, -25°to 20° vertical FOV, 0.02 noise standard deviation, 10 Hz rotation frequency |
| 4 × Radar | 150m range, 120° horizontal FOV, 40° vertical FOV |
| GPS & IMU | heading noise 0.1°, speed noise 0.2m/s |

Table 3: **Comparisons of our dataset and existing dataset.**

| Dataset | Radar (360°) | V2V | V2I | I2I | Camera Frame | LiDAR Frame | Radar Frame |
|---|---|---|---|---|---|---|---|
| OPV2V | | ✓ | | | 44k | 11k | - |
| V2X-Sim | | ✓ | ✓ | | 0 | 10k | - |
| V2XSet | | ✓ | ✓ | | 44k | 11k | |
| V2X-R | | ✓ | ✓ | | 150k | 37k | 37k |
| V2XSet-R(Ours) | ✓ | ✓ | ✓ | ✓ | 142k | 31k | **142k** |

## 5 EXPERIMENTS

### 5.1 EVALUATION AND IMPLEMENTATION DETAILS

Each agent uses a PointPillar encoder. Each Q-DA block uses $H = 4$ attention heads. The $D$-dimensional query vector is evenly divided into 4 channel groups. Each head predicts its own offsets and importance weights, enabling the model to capture complementary spatial patterns.

For every head, the block samples $P = 4$ sparse points. Thus, a single query aggregates at most $H \times M \times P = 4 \times 2 \times 4 = 32$ neighbor features, which proved sufficient to cover the extents of object. A sweep showed that increasing $P$ beyond 4 yielded marginal accuracy gains ($< 0.2\%$ mAP) but noticeably higher latency, so we adopt $\{H=4,\ P=4\}$ as a balanced default. Models are trained for 30 epochs with Adam (learning rate $5 \times 10^{-4}$, weight decay $10^{-4}$), a multi-step LR decay im epochs 10 and 15 ($\gamma = 0.2$).

### 5.2 QUANTITATIVE EVALUATION

Table 4: **The effect of the positional encoding in our proposed fusion module.**

| Positional Encoding | 3D mAP@Validation | | | 3D mAP@Testing | | |
|---|---|---|---|---|---|---|
| | 0.3 | 0.5 | 0.7 | 0.3 | 0.5 | 0.7 |
| Excluded | 79.2 | 75.6 | 52.1 | 83.6 | 80.2 | 61.3 |
| Incorporated | 79.9 | 76.1 | 52.0 | 85.3 | 82.3 | 64.1 |

4D automotive radars return sparse point clouds per frame, making accurate cooperative detection especially challenging. Despite this sparsity, our SparseFusion network achieves the best 3D mAP in both the validation and the public testing splits of the V2X-R benchmark across all IoU thresholds. In the validation dataset, SparseFusion attains 79.9/76.1/52.0% mAP at IoU@0.3,0.5,0.7, improving on the SOTA by 4.3, 5.8, and 4.1 percent, respectively. Similarly, in the test split, SparseFusion still achieves the best performance 85.3/82.3/64.1% mAP. These consistent margins demonstrate that

dual-stage fusion not only compensates for radar sparsity but also scales well with the number of collaborating vehicles, yielding state-of-the-art accuracy.

Table 5: **Performance comparisons.** (a) Number of deformable layers. (b) Ablation of our method.

<table>
<tr><td colspan="7" align="center">(a) Deformable layers</td></tr>
<tr><td rowspan="2">Num of
Deform Layer</td><td colspan="3">3D mAP@Validation</td><td colspan="3">3D mAP@Testing</td></tr>
<tr><td>0.3</td><td>0.5</td><td>0.7</td><td>0.3</td><td>0.5</td><td>0.7</td></tr>
<tr><td>1</td><td>77.2</td><td>73.5</td><td>49.6</td><td>82.0</td><td>78.2</td><td>59.1</td></tr>
<tr><td>3</td><td>79.9</td><td>76.1</td><td>52.0</td><td>85.3</td><td>82.3</td><td>64.1</td></tr>
<tr><td>6</td><td>79.2</td><td>75.5</td><td>51.3</td><td>84.2</td><td>81.1</td><td>63.4</td></tr>
</table>

<table>
<tr><td colspan="5" align="center">(b) Ablation study</td></tr>
<tr><td>Spatial
Fusion</td><td>Deform
Fusion</td><td>MAFE</td><td>AP 0.5</td><td>AP 0.7</td></tr>
<tr><td>✓</td><td></td><td></td><td>77.6</td><td>55.0</td></tr>
<tr><td></td><td>✓</td><td></td><td>81.1</td><td>62.9</td></tr>
<tr><td>✓</td><td>✓</td><td></td><td>81.9</td><td>63.8</td></tr>
<tr><td>✓</td><td>✓</td><td>✓</td><td>82.3</td><td>64.1</td></tr>
</table>

## 5.3 Qualitative Evaluation

When fusing sparse features from multiple agents that observe the same spatial voxel, the information density is often unbalanced because one agent may record only a few sparse radar returns, while another has no returned points, producing an empty voxel and leading to the sparse–to–empty voxel problem, where informative but weak signals are diluted by the empty features. The proposed dual fusion module assigns an adaptive weight to every candidate voxel, allowing the network to dynamically select the most informative agent-specific representation. By emphasizing valuable sparse voxels and suppressing empty or noisy ones, the module strengthens cross-agent interactions without being confused by missing data.

As visualized in Fig. 3, we visualize the extracted features by each agent and the fused features map to show the feature selection process. Red circles mark sparse voxels from agent $i$ that are amplified after fusion, while purple circles mark empty/noisy voxels from agent $j$ that are deliberately discarded. The dashed arrows illustrate how the module routes only the selected information to the fusion layer and ultimately to the detector. This qualitative evidence shows that our dual-fusion strategy successfully mitigates the sparse–to–empty voxel issue, preserves discriminative information from the sparse voxel features and avoid the dilution effect of the empty voxel.

## 5.4 Ablation Study

Table 6: **Range analysis of different fusion methods.** We evaluate 3D multi-agent perception performance across various ranges to compare the effectiveness of each method to fuse sparsely ranged objects.

<table>
<tr><td rowspan="2">Method</td><td colspan="4">3D mAP@0.5</td><td colspan="4">3D mAP@0.7</td></tr>
<tr><td>[0,100)</td><td>[0,30)</td><td>[30,50)</td><td>[50, 100)</td><td>[0,100)</td><td>[0,30)</td><td>[30,50)</td><td>[50, 100)</td></tr>
<tr><td>V2X-ViT Xu et al. (2022)</td><td>73.5</td><td>89.4</td><td>73.9</td><td>65.7</td><td>42.5</td><td>69.1</td><td>41.9</td><td>30.8</td></tr>
<tr><td>Adafusion Qiao & Zulkernine (2023)</td><td>77.8</td><td>91.6</td><td>77.2</td><td>72.3</td><td>55.4</td><td>77.0</td><td>51.2</td><td>49.1</td></tr>
<tr><td>Coalign Qiao & Zulkernine (2023)</td><td>75.7</td><td>89.6</td><td>74.4</td><td>68.1</td><td>52.0</td><td>72.3</td><td>48.2</td><td>40.7</td></tr>
<tr><td>SICP Qiao & Zulkernine (2023)</td><td>65.1</td><td>80.2</td><td>63.2</td><td>59.3</td><td>63.4</td><td>81.6</td><td>58.8</td><td>53.1</td></tr>
<tr><td>Attfusion Xu et al. (2021b)</td><td>74.9</td><td>90.0</td><td>70.9</td><td>68.2</td><td>48.0</td><td>69.9</td><td>40.6</td><td>42.3</td></tr>
<tr><td>Our</td><td>83.0</td><td>93.9</td><td>82.8</td><td>76.1</td><td>64.6</td><td>83.2</td><td>63.4</td><td>57.0</td></tr>
</table>

### 5.4.1 Decomposed Analysis of the Q-DA

Table 5 contrasts three configurations of our fusion pipeline and their AP at IoU thresholds of 0.5 and 0.7. With only the spatial fusion module enabled, the detector attains 77.6 AP@IoU 0.5 and 55.0 AP@IoU 0.7. These scores serve as our reference. Substituting the spatial module with the deformable counterpart raises performance to 81.1 and 62.9, which indicates that adaptive receptive fields capture cross-agent correspondence more effectively than fixed spatial sampling. Activating all modules yields the highest accuracy, reaching 82.3 AP@0.5 and 64.1 AP@0.7. This shows that precise alignment and flexible aggregation are complementary.

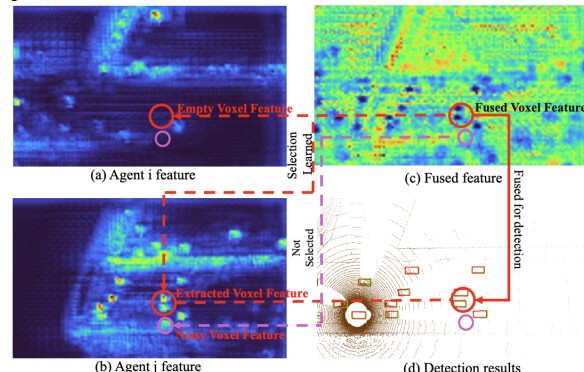

Figure 3: **Visualization of fused features and selection.** (a)–(d) as described. Red circles: strengthened sparse-to-empty voxel interactions; purple circles: noisy voxels discarded.

Table 7: **Comparison of the existing fusion methods on different sensor data.**

| Fusion Module | 4D Radar | | | LiDAR | | |
|---|---|---|---|---|---|---|
| Comparison | IoU=0.3 | IoU=0.5 | IoU=0.7 | IoU=0.3 | IoU=0.5 | IoU=0.7 |
| Self-Att Xu et al. (2021b) | 81.3 | 74.9 | 47.9 | 91.5 | 90.0 | 82.4 |
| Multi-Swin Xu et al. (2022) | 80.6 | 73.5 | 42.6 | 91.2 | 90.0 | 79.8 |
| Multi-Scale Lu et al. (2023) | 81.6 | 75.7 | 52.0 | 91.1 | 90.1 | 83.7 |
| Our | 85.3 | 82.3 | 64.1 | 93.0 | 92.3 | 86.7 |

### 5.4.2 POSITIONAL ENCODING

Table 4 compares the detection performance when positional encoding is excluded or incorporated into the fusion transformer. In the test split, with positional encoding, the 3D mAP increases from 83.6 to 84.2 at IoU 0.3, and from 80.2 to 81.1 at IoU 0.5. The consistent improvements on the larger test set confirm that positional cues make cross-agent fusion more robust and ultimately lead to higher detection accuracy.

### 5.4.3 NUMBER OF DEFORMABLE LAYERS

Table 5 studies how the depth of the deformable attention branch influences the detection accuracy. Starting with a single deformable layer, the model reaches 77.2 and 73.5 mAP on the validation split at IoU thresholds 0.3 and 0.5, and 82.0 and 78.2 mAP on the test split. Increasing the depth to three layers yields the best scores in all metrics. Expanding the stack further to six layers doesn't offer additional benefit. The drop suggests diminishing returns from deeper attention, possibly due to the optimization difficulty. These results indicate that three deformable layers strike a good balance between representation power and efficiency.

### 5.4.4 RANGE ANALYSIS

Objects located far from the ego agent are typically captured by fewer radar returns. Their voxel features become sparse and are often completely empty. Table 6 shows that our fusion strategy achieves the highest accuracy in every distance interval and maintains a clear margin when the targets are beyond 30 meters. The improvement in the far-range bands indicates that the proposed multi-agent dual fusion mechanism can select complementary information from neighboring voxels and recover information that the ego view alone would miss. Consequently, our method yields more consistent performance across distances and strengthens detection of distant sparsely represented objects.

### 5.4.5 COMPARISON WITH EXISTING FUSION METHODS

Table 7 compares our approach with mainstream multi-agent feature fusion methods by using the different sensor measurements as input. Because they calculate the attention scores only among features that occupy exactly the same spatial voxel across agents, such fusion ignores attention score in neighboring voxels and the feature interaction with nearby regions. Our method enlarges this interaction range by learning to select informative features from nearby multi-agent voxels instead of being restricted to exact voxel position. This broader receptive field lets the network fill sparse or empty regions with complementary evidence from other agents, and therefore produces richer fused representations. The numbers in Table 7 confirm the benefit. In the test split our method exceeds Self-Att by 4.0, 7.4, and 16.2 points, Multi-Swin by 4.7, 8.8, and 21.5 %, and Multi-Scale by 3.7, 6.6, and 12.1%.

## 6 CONCLUSION

We introduced **SparseFusion** to overcomes the attention-based affinity calculation constraint and enhance the sparse multi-agent features. By applying query-based deformable attention to sample informative cross-voxel cross-agent locations and augmenting non-empty BEV pillars with RoPE-based sparse self-attention, SparseFusion enables richer context aggregation under sparse sensing. Experiments on both V2XSet-R and V2X-R show that SparseFusion sets a new SOTA in cooperative radar detection (**82.3/64.1** mAP at IoU 0.5/0.7) and further generalizes to occlusion-prone camera BEV features. These results confirm that explicitly modeling cross-voxel evidence is essential for robust multi-agent perception with sparse modalities.

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

# A APPENDIX

## A.1 DATASET SETUP

We conducted experiments using the `V2XR_adafusion` dataset. The training data was stored under `/v2x_radar/new`, and validation was performed on `/v2x_radar/test`. Each training sample included a maximum of five connected autonomous vehicles (CAVs).

## A.2 PREPROCESSING

A sparse voxelization strategy was adopted using the `SpVoxelPreprocessor`. The voxel size was set to $[0.4, 0.4, 10]$, with at most 32 points per voxel. The maximum voxel count was 32,000 during training and 70,000 at test time. Each CAV's point cloud range was limited to $[-140.8, -40, -3, 140.8, 40, 7]$.

## A.3 DATA AUGMENTATION

We applied augmentation in the world coordinate frame, including:

- random flipping along the $x$-axis,
- random rotation in the range $[-0.785, 0.785]$ radians ($\pm 45°$),
- random scaling with factors sampled from $[0.95, 1.05]$.

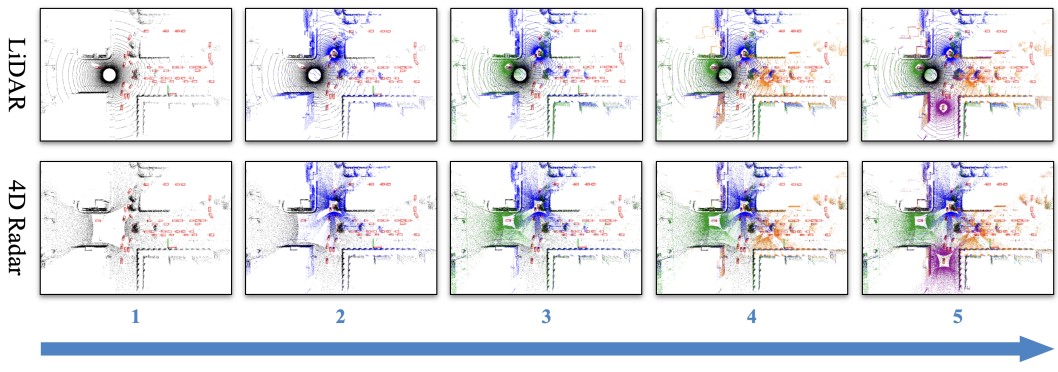

Figure 4: **The visualization of our radar and LiDAR dataset.**

## A.4 MODEL ARCHITECTURE

Our backbone is based on **PointPillars with cooperative spatial fusion**. Voxel features were extracted using a `PillarVFE`, followed by `PointPillarScatter` and a BEV convolutional backbone with three blocks of sizes $[64, 128, 256]$. A shrinking head reduced the 384-dimensional features to 256.

For multi-agent cooperation, we employed the `SpatialFusion_deform` module, which integrates per-agent features via a **multi-modal deformable attention fuser (MFAFuser)**. Specifically, per-agent BEV feature maps are aggregated using both max- and mean-pooled representations, followed by deformable cross-attention layers that align radar and LiDAR features in BEV space. Positional encodings were applied through learnable embeddings to preserve spatial context.

Finally, task-specific heads generated classification and regression outputs using convolutional layers with $1 \times 1$ kernels, predicting anchor-based bounding boxes and objectness scores.

## A.5 DETECTION HEAD AND POSTPROCESSING

Anchor boxes were defined with dimensions 3.9m $\times$ 1.6m $\times$ 1.56m and two orientations ($0°$ and $90°$). Positive and negative sample thresholds were 0.6 and 0.45, respectively. A score threshold of 0.20 was used at inference. Non-maximum suppression employed a threshold of 0.15, with a cap of 100 objects per frame.

## A.6 LOSS FUNCTION

We adopted the `PointPillarLoss`, combining classification and regression objectives, weighted by 1.0 and 2.0, respectively.

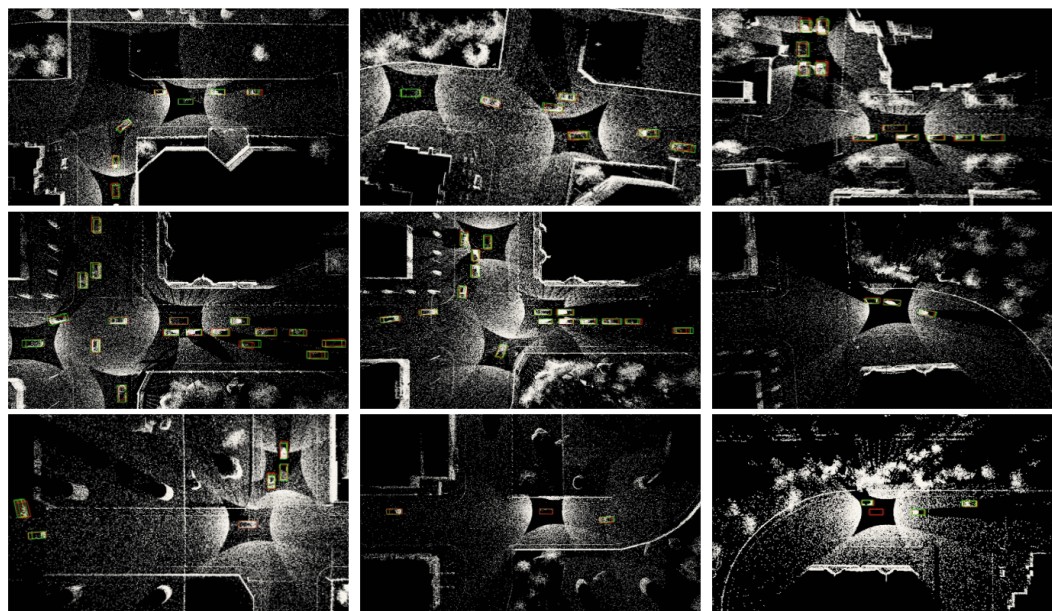

Figure 5: **The visualization of the radar detection results of our multi-agent perception framework.**

### A.7 OPTIMIZATION AND SCHEDULE

The model was trained with the Adam optimizer, using a learning rate of $5 \times 10^{-4}$, weight decay of $10^{-4}$, and $\epsilon = 10^{-10}$. A multi-step learning rate scheduler was applied, reducing the learning rate by a factor of 0.2 at epochs 10 and 15.

### A.8 TRAINING PROTOCOL

Training was performed for 50 epochs with a batch size of 1. Model evaluation and checkpointing were conducted after every epoch.

## B VISUALIZATION OF V2XSET-R DATASET AND PERCEPTION RESULTS

To better understand the characteristics of the proposed dataset, we provide a visualization of the **V2XSet-R** dataset in Figure 4. The dataset contains synchronized LiDAR and 4D radar data collected from multiple connected and autonomous vehicles (CAVs).

Each column in the figure corresponds to one participating agent, ranging from one to five. The top row shows the LiDAR point cloud representations, while the bottom row illustrates the corresponding radar point clouds. LiDAR provides dense geometric structure of the scene, whereas radar contributes complementary information with robustness to adverse weather conditions and long-range detection.

As the number of cooperating agents increases, we can clearly observe a progressive enrichment of the scene representation. Single-agent perception suffers from occlusion and limited field of view, but multi-agent collaboration significantly enhances spatial coverage and object visibility. This highlights the importance of cooperative perception for robust and reliable V2X perception systems.

Figure 5 illustrates the radar-based detection results of our proposed multi-agent perception framework. Each subplot corresponds to a different driving scenario, where radar point clouds are visualized in the background and detected objects are highlighted with red bounding boxes. The results demonstrate that our framework can effectively localize vehicles and surrounding objects under diverse and complex urban layouts. Notably, the detections remain reliable even in regions with sparse or noisy radar returns, showing the robustness of radar sensing when integrated into cooperative multi-agent perception. These visualizations confirm that our multi-agent fusion significantly enhances the multi-agent feature interaction.

