# OpenReview forum: "Perception Through Sparsity: Fusing and Enhancing Multi-Agent Sparse Representation with Deformable Cross-Attention"
_ICLR.cc/2026/Conference — ICLR 2026 Conference Withdrawn Submission_

### Official Review · Reviewer_mZKS · 2025-10-31

**Soundness:** 3
**Presentation:** 2
**Contribution:** 2
**Rating:** 4
**Confidence:** 4

**Summary:**

This paper targets the challenge that multi-agent feature interactions are prone to sparsity, and is reasonably motivated by the premise that full field-of-view point clouds from 4D radar can strengthen V2X perception. Building on this motivation, the method incorporates several novel design choices and demonstrates a degree of innovation. However, the empirical evaluation appears fragmented and insufficiently comprehensive, and the exposition would benefit from significant refinement to improve clarity and rigor.

**Strengths:**

1. Clear, consequential problem framing. The paper tackles a concrete challenge—sparsity in multi-agent feature interaction—leveraging radar’s inherent robustness under adverse weather and low-light conditions. In light of the scarcity of broad, full-FOV data for radar-based collaborative perception, the authors introduce V2XSet-R, a large-scale dataset with 360° coverage, which constitutes a meaningful contribution to the community.
2. Well-motivated methodology with novel fusion design. The motivation is clear and logically developed. The proposed dual-fusion strategy removes the constraint that features must align at identical voxel locations across agents, and the added feature-enhancement module effectively densifies sparse multi-agent features.
3. Sound empirical validation. The paper includes component-wise ablations that substantiate the effectiveness of each proposed module.

**Weaknesses:**

1. Inadequate baselines and a confusing Table 1. Table 1 is difficult to interpret and omits essential comparisons to established methods such as CoBEVT [1]. On the V2X-R radar-based detection benchmark, CoBEVT reports 85.74 mAP at IoU = 0.3; without head-to-head results (and comparisons to other recent radar-centric works, e.g., V2X-R [2]), the claimed methodological advances are not convincingly validated. Results on the proposed V2XSet-R are also overly minimal and lack sufficient comparisons to prior art.
2. Disorganized presentation and unclear attribution in experiments. The experimental section is poorly structured: Table 1 appears in an unusual location and is not referenced anywhere in the text. The paper claims SparseFusion achieves 79.9/76.1/52.0% mAP, exceeding “SOTA” by 4.3/5.8/4.1, yet the specific SOTA baselines are not identified. Multiple tables also fail to state which dataset each result pertains to, obscuring the setup and hindering interpretability and reproducibility.
3. Questionable radar point-cloud density in V2XSet-R. Visualizations suggest that the radar point clouds in V2XSet-R are substantially denser than real radar observations. A quantitative analysis is needed to verify this phenomenon and to reconcile it with the paper’s central claim of addressing feature sparsity.
References:
[1] Xu et al., CoBEVT: Cooperative Bird’s Eye View Semantic Segmentation with Sparse Transformers, 2022.
[2] Huang et al., V2X-R: Cooperative LiDAR-4D Radar Fusion with Denoising Diffusion for 3D Object Detection, CVPR 2025.

**Questions:**

1. Baselines & reporting clarity. Can you add head-to-head results against strong radar-centric baselines—e.g., CoBEVT [1] and V2X-R [2]—on both the official V2X-R radar benchmark and your V2XSet-R, with aligned settings (same splits, training budget, IoU thresholds such as 0.3)? Please also (i) explicitly reference Table 1 in the text, (ii) state the dataset for every table, and (iii) name the exact “SOTA” methods behind the claimed +4.3/+5.8/+4.1 mAP gains (79.9/76.1/52.0).

2. V2XSet-R realism & sparsity. Visualizations suggest V2XSet-R radar point clouds are denser than real 4D radar. Could you provide a quantitative validation (per-frame point counts, range–azimuth–Doppler resolution, SNR distributions, density vs. range) and compare these statistics to real-sensor reports? Additionally, include a sensitivity study that progressively sparsifies the radar input to show the dual-fusion and densification modules still yield gains under realistic sparsity.

---

### Official Review · Reviewer_DwSK · 2025-10-31

**Soundness:** 3
**Presentation:** 3
**Contribution:** 3
**Rating:** 6
**Confidence:** 3

**Summary:**

This paper introduces V2XSet-R, the first large-scale dataset providing complete 360° radar coverage from both vehicles and infrastructure. Building on this, the authors propose SparseFusion, a dual-stage fusion framework tailored for sparse multi-agent perception. Experiments on V2XSet-R and V2X-R show that SparseFusion achieves state-of-the-art 3D mAP and generalizes well to camera-based BEV features, demonstrating its effectiveness in robust multi-agent perception with sparse modalities.

**Strengths:**

1. Novel Dataset Contribution: V2XSet-R fills a critical gap in radar-based V2X benchmarks by providing complete 360° radar coverage from vehicles and infrastructure, along with large-scale data (142k radar frames, 160k annotated 3D boxes).

2. Innovative Fusion Design: The dual-stage fusion (spatial fusion + query-based deformable cross-attention) effectively addresses sparsity issues in radar data.

3. Strong Experimental Validation: Comprehensive experiments (quantitative, qualitative, ablation) confirm SparseFusion’s superiority.

4. Modality Agnosticism: SparseFusion generalizes to LiDAR and camera BEV features, not just radar, increasing its practical utility for multi-modal V2X perception systems.

**Weaknesses:**

1. Limited Real-World Data: V2XSet-R is simulation-based (using OpenCDA + CARLA), lacking real-world radar data.

2. Computational Complexity Details Missing: While the paper mentions efficiency, it lacks explicit analysis of computational cost (e.g., FLOPs, inference time) compared to baselines. For V2X systems with strict latency constraints, this information is critical to assess practical feasibility.

3. Lack of Comparison to Recent Methods: Some recent  cooperative perception works (e.g., HEAL [r1], Heteocooper [r2]) are not included in comparisons.

[r1] An Extensible Framework for Open Heterogeneous Collaborative Perception
[r2] Hetecooper: Feature collaboration graph for heterogeneous collaborative perception

**Questions:**

See the weaknesses.

---

### Official Review · Reviewer_yCwX · 2025-11-04

**Soundness:** 2
**Presentation:** 2
**Contribution:** 2
**Rating:** 2
**Confidence:** 4

**Summary:**

This work makes two key contributions: (i) it introduces a large-scale dataset that provides 360° radar sensor coverage, offering a valuable foundation for the community to advance research on radar-based cooperative perception; and (ii) it proposes SparseFusion, a novel cooperative perception framework motivated by the need to aggregate cross-voxel information across BEV locations. Experimental results show that SparseFusion is both effective.

**Strengths:**

1. This work addresses a common issue in cooperative perception, about how to effectively leverage spatial context during the feature fusion stage, and proposes a novel solution based on deformable attention.

2. The paper makes two-part contributions: (1) the introduction of a new large-scale collaborative perception dataset, and (2) a dual-stage feature fusion method that enhances cross-agent interaction under sparse sensing conditions.

**Weaknesses:**

1. Missing comparison with relevant methods. Please discuss and compare your approach with CoBEVT [1], which also aggregates global spatial context from all agents, both in terms of methodology and experimental performance.

2. Experimental setup and analysis require more detail. In Section 5.2, it is unclear what the compared “SOTA” method refers to. The authors should explicitly specify which baseline or prior work is used for comparison. In addition, the dataset used for evaluation should be clearly indicated (e.g., whether the results are obtained on V2XSet-R or another benchmark).

3. Limited evaluation scope. The evaluation range appears narrow. It would strengthen the paper to include results on publicly available collaborative perception datasets (e.g. OPV2V[2], DairV2X[3]) to better demonstrate the generalizability of the proposed approach.

4. Writing and presentation quality should be improved. Several expressions could be refined for clarity and formality. For example, the use of “addresses” in Line 255 is awkward; the phrase “enter” in Line 211 should be revised to “be fed into”; and the unnecessary line break in Equation (2) should be removed.

5. Method description lacks clarity. The explanation of the “sparse sampling with deformable attention” process is confusing. The authors should better clarify the overall input flow and model structure. For instance, in Equation (5), the query seems to be obtained by summing across attention heads, but in the following equation, q^out is derived by concatenating the outputs from each head. This inconsistency needs further explanation.

[1] Xu, R., Tu, Z., Xiang, H., Shao, W., Zhou, B., & Ma, J. (2022). CoBEVT: Cooperative bird's eye view semantic segmentation with sparse transformers. arXiv preprint arXiv:2207.02202.

[2] Xu, Runsheng, et al. "Opv2v: An open benchmark dataset and fusion pipeline for perception with vehicle-to-vehicle communication." 2022 International Conference on Robotics and Automation (ICRA). IEEE, 2022.

[3] Yu, Haibao, et al. "Dair-v2x: A large-scale dataset for vehicle-infrastructure cooperative 3d object detection." Proceedings of the IEEE/CVF conference on computer vision and pattern recognition. 2022.

**Questions:**

1. How is the q^{out} be computed? The description is different in equation 5 and followups.

---

### Official Review · Reviewer_rdRY · 2025-11-07

**Soundness:** 2
**Presentation:** 3
**Contribution:** 2
**Rating:** 2
**Confidence:** 4

**Summary:**

This paper proposes (1) V2XSet-R, a large-scale dataset providing 360° radar coverage for V2X perception (including V2V, V2I, and I2I settings), and (2) SparseFusion, a dual-stage sparse fusion mechanism that overcomes sparsity-induced degeneration and enhances cross-agent feature interaction through query-based deformable attention.

**Strengths:**

This paper has twofolds contributions.

S1: Introduces a 360° radar V2X dataset, enabling more comprehensive study of cooperative radar perception.

S2: Proposes an effective sparse fusion design that notably improves cross-agent feature aggregation and performance under sparse sensing.

**Weaknesses:**

W1: The dual fusion module in the proposed sparsefusion mechanism shares a bit similar idea of paper "Align before Collaborate: Mitigating Feature  Misalignment for Robust Multi-Agent Perception [ECCV2024]" which limits its novelty. It would be better to clearly articulate the differences and include comparison.

W2: While the newly proposed dataset is valuable, it remains simulation-based. Given the availability of real-world datasets such as V2XScenes [ICCV 2025] that already include V2V/V2I/I2I settings, and the ease of extending V2X-R to achieve 360° radar coverage, the dataset’s contribution appears limited in originality.

**Questions:**

Please refer to the weaknesses outlined above. Given above reasons, I would give rating of 3 if there is an option. However, I would raise my score if the authors adequately address the concerns highlighted.

---

### Note · Authors · 2025-12-30

I have read and agree with the venue's withdrawal policy on behalf of myself and my co-authors.